# Optical Investigation of Action Potential and Calcium Handling Maturation of hiPSC-Cardiomyocytes on Biomimetic Substrates

**DOI:** 10.3390/ijms20153799

**Published:** 2019-08-03

**Authors:** Josè Manuel Pioner, Lorenzo Santini, Chiara Palandri, Daniele Martella, Flavia Lupi, Marianna Langione, Silvia Querceto, Bruno Grandinetti, Valentina Balducci, Patrizia Benzoni, Sara Landi, Andrea Barbuti, Federico Ferrarese Lupi, Luca Boarino, Laura Sartiani, Chiara Tesi, David L. Mack, Michael Regnier, Elisabetta Cerbai, Camilla Parmeggiani, Corrado Poggesi, Cecilia Ferrantini, Raffaele Coppini

**Affiliations:** 1Department of Experimental and Clinical Medicine, Division of Physiology, Università degli studi di Firenze, 50134 Florence, Italy; 2Department NeuroFarBa, University of Florence, 50134 Florence, Italy; 3European Laboratory for Non-Linear Spectroscopy (LENS), 50019 Florence, Italy; 4National Institute of Optics, CNR-INO, 50125 Florence, Italy; 5Department of Biosciences, Università degli studi di Milano, 20137 Milan, Italy; 6Istituto Nazionale di Ricerca Metrologica INRiM, 10129 Turin, Italy; 7Department of Rehabilitation Medicine, University of Washington, Seattle, WA 98108, USA; 8Department of Bioengineering, University of Washington, Seattle, WA 98108, USA; 9Department of Chemistry “Ugo Schiff”, University of Florence, 50134 Florence, Italy

**Keywords:** human induced pluripotent stem cells, cardiomyocytes, fluorescence, maturation, action potential, calcium handling, hydrogels, long-term culture

## Abstract

Cardiomyocytes from human induced pluripotent stem cells (hiPSC-CMs) are the most promising human source with preserved genetic background of healthy individuals or patients. This study aimed to establish a systematic procedure for exploring development of hiPSC-CM functional output to predict genetic cardiomyopathy outcomes and identify molecular targets for therapy. Biomimetic substrates with microtopography and physiological stiffness can overcome the immaturity of hiPSC-CM function. We have developed a custom-made apparatus for simultaneous optical measurements of hiPSC-CM action potential and calcium transients to correlate these parameters at specific time points (day 60, 75 and 90 post differentiation) and under inotropic interventions. In later-stages, single hiPSC-CMs revealed prolonged action potential duration, increased calcium transient amplitude and shorter duration that closely resembled those of human adult cardiomyocytes from fresh ventricular tissue of patients. Thus, the major contribution of sarcoplasmic reticulum and positive inotropic response to β-adrenergic stimulation are time-dependent events underlying excitation contraction coupling (ECC) maturation of hiPSC-CM; biomimetic substrates can promote calcium-handling regulation towards adult-like kinetics. Simultaneous optical recordings of long-term cultured hiPSC-CMs on biomimetic substrates favor high-throughput electrophysiological analysis aimed at testing (mechanistic hypothesis on) disease progression and pharmacological interventions in patient-derived hiPSC-CMs.

## 1. Introduction

Cardiomyocytes generated from human inducible pluripotent stem cells (hiPSCs) have become a powerful tool for the basic-science community dedicated to heart regeneration, high-throughput drug screening, cardiac genetics and disease modelling.

For example, the breakthrough of reprogramming somatic cells from patients into hiPSCs has opened the possibility to characterize specific genotype/phenotypes for both the patient and their progeny. Thus hiPSC-derived cardiomyocytes (hiPSC-CMs) can serve as a unique platform for unveiling novel molecular targets of mechanisms underlying inherited cardiomyopathies. However, since the inception of hiPSC-CMs [1], a major focus has been to produce cells that mirror mechanisms found in end-stage human samples or animal models. Only a few studies have described the time course of disease pathophysiology and/or novel mechanisms for potential therapeutic strategies [2]. It is widely accepted that post-differentiated hiPSC-CMs are immature in both structure and function [3]. “Immaturity” comprises a wide spectrum of fetal-type features, such as automaticity (spontaneous contraction), gene expression patterns [4] including ion channels [5], and electrophysiological signals and contractile properties at whole cell [6] and myofibril level [7,8]. Using current approaches, hiPSC-CMs display developmental hallmarks clearly discernable from adult cardiomyocytes isolated from fresh cardiac tissue, thus providing a useful model to study the progression of disorders that reflect cardiac diseases at early-stage. Therefore, much more work is still needed to categorize how the functional properties of hiPSC-CMs change over time in culture to avoid misleading conclusions for normal and disease phenotypes.

There is a significant need to standardize methods and techniques so that results from different laboratories are sufficiently comparable in terms of the developmental stage of hiPSC-CMs. Many strategies have emerged to improve hiPSC-CM structure/function maturation [3]. Among these, we previously reported that hiPSC-CMs benefit from long-term culture on micropatterned surfaces. In later-stages (80–100 days post differentiation), healthy hiPSC-CMs exhibited a dramatic increase of the cell aspect ratio and myofibril alignment, with evidence of caveolae or emerging transverse (T) tubules and expression of sarcomere protein isoforms that are generally not detected at earlier post-differentiation time points [7,9].

In earlier times post-differentiation hiPSC-CMs exhibit spontaneous contractile activity due to the absence or very low expression of the inward-rectifier potassium channels (I_K1_) that stabilizes the membrane potential and the presence of the pacemaker funny current (I_f_) [10]. T-tubules have been rarely described [11] and the sarcoplasmic reticulum (SR) is still poorly developed. In this scenario, sarcolemma calcium entry is the main source for contraction and the sodium calcium exchanger (NCX) is mainly responsible for intracellular calcium removal [12]. Consequently, calcium handling kinetics are slower in hiPSC-CMs compared to adult cardiomyocytes [13].

Simultaneous optical recordings of action potential and calcium transients were previously reported from cell monolayers [14] or using genetically encoded reporters [15] at undefined maturation stages. In this work, we present a systematic functional characterization of action potential and calcium handling maturation using a dual optical method at specific time points in long term hiPSC-cardiomyocytes on hydrogel-based micropatterned substrates. Hydrogel-based substrates are easy to handle and to shape with nanoscale topographies using soft lithography techniques. Their stiffness can be tuned to mimic the native cardiac extracellular matrix. A simultaneous optical recording approach of action potential and calcium transient is presented here, to correlate these two parameters and describe developmental changes of later-stages (days 60–75–90) in the development of individual hiPSC-CM excitation contraction coupling (ECC).

We report that only in later-stages do hiPSC-CMs respond to field stimulation at various pacing frequencies and to inotropic agents, likely resembling calcium handling features of human adult cardiomyocytes [13]. This approach can be applied to investigate time-dependent events of more mature hiPSC-CMs and modelling early (vs. late) morpho/functional cardiomyocyte changes associated with inherited cardiomyopathies. This approach may allow the identification of molecular targets for the development of novel “disease”-specific, or even “disease stage”-specific, therapeutic approaches, and potentially prevent the pathogenesis of cardiomyopathies before the clinical manifestation [16].

## 2. Results

### 2.1. Simultaneous Recordings of Action Potential and Calcium Transient from Single hiPSC-Cardiomyocytes

Human iPSC were differentiated into cardiomyocytes (hiPSC-CMs) using a monolayer-directed differentiation protocol as previously described [17]. Briefly, a monoloayer with about 70% of confluency of hiPSC colonies was exposed to the medium A (day 0) for two days to initiate the differentiation process. Spontaneous beating monolayers are visible in about 8–10 days after cardiac induction and employed for experiments at three “later-stages” time points: day 60, 75 and 90 (Figure 1A). At day 20 after differentiation, beating monolayers were selected and digested into single cells and plated at a cell density of 20,000 cell/cm^2^ onto fibronectin-coated custom-made polyethylene glycol-diacrylate (PEG-DA) hydrogel-based surfaces with a microstructured topography. The geometry of this biomimetic substrate is reported in Figure 1B, presenting an array of parallel micro-grooves with 0.6 µm width, 1.5 µm height and 1.4 µm distance between lines. Of note, the stiffness (Young modulus) of the hydrogel-based surfaces was in the order of 100 kPa, less compared to our previous reports, which was 6.7 MPa [7] and was chosen in order to more closely mimic physiological conditions of the human cardiac extracellular matrix [18]. In order to perform a systematic functional characterization, we developed a custom-made experimental setup based on an epifluorescence microscope to simultaneously record action potentials and calcium transients using fluorescent indicators (FluoVolt and Cal630, respectively, Figure 1C, Appendix A).

### 2.2. Action Potential and Calcium Transient of Human iPSC-Versus Adult Cardiomyocytes

At later stages of maturation, single rod-shaped hiPSC-CMs are responsive to electrical field stimulation at various pacing rates. Action potentials (AP) was measured by acquiring the voltage-sensitive fluorescent dye (Fluovolt) signal at day (d) 75 and 90 (Figure 2A). At day 75, AP duration (APD) did not show a significant rate adaptation (APD50 at 1 and 2 Hz were comparable). At d90, instead, APD50 was shortened at 2 Hz compared to 1 Hz, highlighting that mature hiPSC-CMs acquired a typical rate-dependent APD adaptation (*p* < 0.01, Figure 2A). Moreover, at day 90 APD50 at 1 Hz was longer, compared to that recorded at day 75 (*p* < 0.01). Simultaneously, calcium transients were measured at each time point (Figure 2B). At both 1 and 2 Hz, the kinetics of calcium rise (time to peak, TTP) and decay (ratio between TTP and time to 50% of CaT decay, RT50) became more rapid with maturation (Figure 2C) (*p* < 0.01) (Appendix A
*hiPSC-CMs no stimulation*, Appendix A 1 Hz, Appendix A 2 Hz). Similarly, calcium transient amplitude was similar at day 60 and 75, but it increased by 50% at d90 (*p* < 0.01 vs. d60, Figure 2C). This behavior closely matches the features of human adult cardiomyocytes (hAdult-CMs, Figure 2D). When APD50 was plotted against calcium transient parameters recorded from individual cells at day 90, Pearson’s correlation coefficient (*r*^2^) estimated from the linear fitting displayed no correlation between APD50 and CaT amplitude. However, there was positive correlation between APD50 and calcium transient RT50. This suggests that cardiomyocytes, which show longer APD, also have a longer calcium transient duration (Figure 3A). The same correlation was found in adult ventricular cardiomyocytes from non-failing non-hypertrophic patients, in which AP and calcium transients were simultaneously measured with standard techniques (Figure 3B).

To validate the reproducibility of functional data from cells of a single cell line, we evaluated the variability among different batches of the same cell line obtained from distinct differentiation runs (p, indicates the passage in culture). CaT amplitude and RT50 values from 2 different cell batches at d60 (Appendix A, respectively) and d90 (Appendix A, respectively) were compared, in order to demonstrate that: (1) the inter-batches variability is comparable to the variability within each individual batch, (2) the maturation of calcium transients over-time observed in each individual batch, reflects that of the general combined population. Appendix A show comparison of CaT amplitude and decay (RT50, ms) among individual differentiations (from a single passage, p) versus the average of total differentiations at day 60 and 90.

However, the increase of CaT amplitude is consistent for individual differentiations at d90 compared to d60. Similarly, distribution of RT50 values were more consistent within single batches at all considered time points (Appendix A).

### 2.3. Sarcoplasmic Reticulum Contribution to Calcium Handling of Later-Stages hiPSC-Cardiomyocytes

The sarcoplasmic reticulum (SR) is the main store of calcium in cardiomyocytes. SR proteins, including RyR2, SERCA-2a and phospholamban (PLB) were shown to be expressed in hiPSC-cardiomyocytes [19]. However, fewer reports have described their functional contribution at later-stages hiPSC-CMs. A post rest potentiation protocol was applied to evaluate the maturation of cardiomyocyte inotropic reserve when cells are cultured in our conditions. hiPSC-CMs were paced at 2 Hz and, following a stimulation pause of 5 s, the amplitude of the first CaT after stimulation is resumed the pause was measured to evaluate the rest-mediated potentiation. This potentiation is related to prolonged SR refilling and enhanced RyR2 channel availability, because of the complete recovery from inactivation/adaptation of RyR channels during the longer rest period (Figure 4A). CaT amplitude after the rest pause displayed a modest potentiation over the amplitude before the pause at day 60. However, the amplitude of post-rest CaTs almost doubled at day 90, suggesting increased SR loading capacity during maturation (day 60 vs. 90, *p* < 0.01). To verify the contribution of SR calcium content, we evaluated caffeine-induced calcium transients (quick exposure to 10 μM caffeine) after a pacing train of 2 Hz. The amplitude of caffeine-induced CaT at d60 was doubled compared to corresponding steady state calcium transients at 2 Hz pacing.

We then measured the amplitude of caffeine-induced CaT estimated as the ratio of caffeine-CaT amplitude (CaT_A CAFF_) and the CaT amplitude before caffeine exposure (CaT_A 2Hz_). In day 60-hiPSC-CMs the relative caffeine-induced CaT amplitude was higher compared to that of hAdult-CMs (Figure 4B). These results indicate a significant contribution of calcium release from the SR to the amplitude of calcium transients starting from day 60 and suggest that the enhanced SR calcium load is the main determinant of the progressive increase of CaT amplitude of hiPSC-CMs from day 60 to day 90.

In addition, the time constant of the decay of caffeine-induced transients (τ, s^−1^) is an indirect measurement of calcium removal operated by the sodium calcium exchanger (NCX) during exposure to caffeine. The time constant of caffeine transient decay was about 25% slower compared to that calculated in hAdult-CMs. This supports the idea that after maturation in our conditions, d60 hiPSC-CM feature calcium removal pathways (e.g., NCX) with a rate near to but still not the same of hAdult-CMs. Moreover, simultaneous recording of AP and calcium dynamics was employed to assess the potentiation of CaT amplitude and the prolongation of CaT decay after the pause, as well as the impact of prolonged pauses on the APD50 of the first beat after the resting period (Figure 4C). Both RT50 of CaT and APD50 of post rest beats were prolonged when compared with steady state beats, at every maturation time point.

Additionally, the duration of post-rest APs and CaTs in the different cells were directly correlated. However, compared to earlier phases (d60), both post-rest APD50 and RT50 became shorter at d90 (Figure 4E,F, respectively), likely suggesting improved cytosolic calcium extrusion via NCX and/or reuptake by SERCA.

### 2.4. Ca-Handling Response to Inotropic Intervention of Later-Stages hiPSC-Cardiomyocytes

Another key aspect we explored during hiPSC-CM maturation is the response to inotropic agents. We tested the effect of 1 μM isoproterenol (ISO) and forskolin (FSK) to mimic the β-adrenergic pathway. Isoproterenol binds β-adrenergic receptor (βAR) type 1 and 2 in cardiac cells and activates the cAMP-PKA phosphorylation pathway. Similarly, FSK stimulates adenylate cyclase rising cAMP levels to activate PKA. We exposed day 90-hiPSC-CMs to ISO or FSK after basal recording of APs and CaTs to evaluate the relative inotropic and lusitropic effect (%) during regular stimulation. ISO reduced APD50 in hiPSC-CMs paced at 1 Hz (Figure 5A).

Similarly, APD50 reduction under ISO was observed in hAdult-CMs (Figure 5B). Accordingly, RT50 of CaT was faster under both ISO and FSK exposure in hiPSC-CMs (Figure 5C), consistent with typical βAR-mediated positive lusitropic effect. In addition, both ISO and FSK enhanced CaT amplitude, indicating positive inotropic response (Figure 5D).

### 2.5. Membrane Potential and Spontaneous Action Potential of Earlier-Stage hiPSC-CMs

To support and validate optical AP measurements, the patch clamp technique was employed to measure APs in early-stage hiPSC-CMs (day 20, 30, 60 p.d.), during stimulation with short current pulses in the current-clamp configuration at different frequencies (0.5–1–2 Hz, Figure 6A). Figure 6B shows that, compared to the earliest-stage (d20), hiPSC-CMs at later stages of maturation showed progressively more negative resting membrane potentials (RMP, mV), larger AP amplitude (mV) and prolongation of AP duration (APD_50_). Moreover, hiPSC-CMs at later-stages (d60) showed a trend towards rate-adaptation of AP duration. However, a clear adult-like rate-dependency of APD develops only at day 90, as shown in Figure 2A. Notably, spontaneous beating frequency, measured in patched cells without stimulation, showed a progressive reduction from day 20 to 60 p.d. (Figure 6C). Of note, mature hiPSC-CMs (from day 60) do not show a significant spontaneous beating activity.

## 3. Discussion

Twelve years after the first study describing the generation of iPSCs [1], there are high expectations related to the use of human hiPSC-CMs from patients or donors as disease models for studying the pathophysiology of cardiac diseases or for testing novel targeted therapies. However, hiPSC-CMs must be extensively validated against native functional human myocardium, in order to define current and potential applications of this widely used technology. Unfortunately, an accurate comparison of the functional features of hiPSC-CMs and native human ventricular cardiomyocytes is still lacking. In this work, we presented a systematic functional characterization of action potential (AP) and calcium handling maturation using a dual optical method at specific time points in later stages hiPSC-CMs (day 60, 75 and 90 post differentiation) cultured on biomimetic substrates with micropatterned topography and physiological stiffness in the range of kPa. Functional data from hiPSC-CMs are compared with those obtained in freshly isolated cardiomyocytes from surgical ventricular samples of non-failing non-hypertrophic patients, collected using standard electrophysiological and fluorescence techniques.

We performed simultaneous recordings of action potential and calcium transients with fluorescent indicators at single cell level, to assess the key regulatory mechanisms of cardiac contraction. Thanks to the use of a fast high-sensitivity and high-resolution camera, our methods allows high throughput screenings of multiple cells in simultaneous, increasing the capability of assessing the variability of functional data among different cells and different runs of differentiation. With our approach, we were able to collect data from hundreds of single hiPSC-CMs in each condition and at each differentiation step, eliminating biases related to selection and small sample size.

The voltage-sensitive dye Fluovolt has been previously shown to exhibit much higher signal fidelity to traditional voltage-clamp recordings, as compared to other commonly used voltage-sensitive dyes, such as ANNEP dyes [20]. As reported before, the amplitude of voltage signals recorded with optical methods is influenced by cell membrane surface area and by the degree of dye loading, hence, this parameter can be used only to assess relative variations of the same cell. Contrarily, the duration of action potentials recorded with optical methods is not affected by cell density, membrane extension or dye concentration [21]. In our hands, the duration of APs recorded from individual hiPSC-CMs using the patch-clamp technique (earlier-stages) and Fluovolt approach (later-stages) were consistently prolonged at any progressive time point. Moreover, the duration of APs recorded with Fluovolt in hiPSC-CMs at 90 days was similar to that recorded in human adult ventricular cardiomyocytes (hAdult-CMs) recorded using traditional voltage-clamp.

Previous findings on long-term cultured hiPSC-CMs have revealed changes of ion current density and properties during cell maturation protocols with prolonged culture time [22]. After long-term maturation on micropatterned hydrogels, hiPSC-CMs showed a prolongation of action potentials, likely associated with a larger contribution of I_CaL_ during the plateau phase, which prevails over the increase of repolarizing outward potassium currents (I_Kr_ and I_to1_), as previously described [23]. Moreover, the more negative resting membrane potential recorded by patch clamp after day 60 implies an increase of ion currents that contribute to the diastolic potential (in particular I_K1_). The increase of I_K1_ may also account for the decrease of spontaneous beating rate at 60 days p.d. and the consequent increased response to external electrical pacing (field stimulation) and frequency changes. Other possible contributors to the acquisition of this physiological behavior during hiPSC-CM maturation are the decrease of funny current (I_f_) and the increase of the delayed rectifier potassium currents (I_Ks_ + I_Kr_) [24]. Many ion currents (e.g., I_to1_) undergo developmental changes, similar to those occurring during the fetal cardiac development [22]. For this reason, the AP profile of early hiPSC-CMs maintains some of the features of fetal cardiomyocytes. However, at day 90 the average AP duration and calcium transients (CaT) duration measured in hiPSC-CMs more closely recapitulates the AP profile and CaT kinetics of ventricular hAdult-CMs. In line with hAdult-CMs, AP duration was correlated with CaT duration in hiPSC-CMs. On the other side, acceleration of CaT duration and adaptivness of CaT kinetics and amplitude to frequency changes at later-stages are likely attributable to increase of SERCA/PLB function, associated with the development of sarcoplasmic reticulum. Of note, the duration of CaT decay in hiPSC-CMs was in the same range of that measured in hAdult-CMs. The presence of a more mature SERCA function is also supported by observation of the caffeine transient decay rate (τ) in hiPSC-CMs, which is near but still not the same compared with hAdult-CMs, suggesting improved NCX function. Not least, we found positive lusitropic effect under β-adrenergic stimulation (ISO and FSK), which entails amplified SERCA contribution (mediated by PLB phosphorylation), paralleled by PKA-induced reduction of APD (I_Ks_ phosphorylation) under adrenergic stimulation. Similarly, CaT rise (time to peak) was in the same range of hAdult-CMs [11].

Sarcoplasmic reticulum (SR) contribution was assessed both by applying stimulation pauses and by estimating SR Ca^2+^load from measurements of caffeine-induced CaT amplitude. Greater SR contribution appeared as the major contributor to the increased CaT amplitude and post rest potentiation during hiPSC-CM maturation. In our conditions, both post rest CaTs and post-rest APs were prolonged as compared with steady state beats. The prolongation of the post rest CaT decay and APD declined in later stages of maturation, supporting the idea of increased role of SERCA and NCX function contributing to calcium removal. Moreover, a more rapid calcium-dependent inactivation of I_Ca-L_ in later stages may account for shorter APD50 after rest. In native cardiomyocytes, a calcium/calmodulin (CaM) dependent inactivation (CDI) of I_Ca-L_ tunes the open probability of this channel [25]. Thereby, later-stages hiPSC-CMs may have increased CaM tethering to the channel C-terminus altering the interaction of CaM with the calcium channel C-terminus accelerating the inactivation. After the resting pause, with larger Ca releases stronger CDI may limit integrated I_Ca_ influx, shortening the APD and further limiting Ca entry and the earlier repolarization, which can favor additional Ca extrusion via NCX [26]. On the other side, faster calcium transient decay after rest may likewise suggest reduced myofibril calcium buffering during the CaT decay phase (faster thin filament de-activation) [27], a likely consequence of the developmental shift from the slow skeletal (fetal) to the cardiac (adult) troponin complex [7,28,29], which can contribute to SR-calcium pump via SERCA. Finally, we report hiPSC-CMs showing a high rate of positive inotropic and lusitropic responses in response to neurohormonal agonists. Previous observations provided evidence of the progressively increased expression of β_1_, β_2_ and β_3_ adrenergic receptors (ARs) over a period of 90 days in culture, showing that β2-ARs are the most prominent source of cAMP/PKA signaling and target of isoproterenol in hiPSC-CMs [30]. We confirmed the presence of a working downstream pathway of adrenergic signalling both via direct AR activation (isoproterenol) and by directly increasing intracellular cAMP levels (FSK). Major targets of PKA-mediated phosphorylation involved in calcium dynamics include I_Ca-L_ channels (inotropic), RyR2 channels (inotropic and chronotropic), PLB (inotropic and lusitropic), cardiac Troponin I (cTnI) (lusitropic) and I_Ks_ (AP shortening, lusitropic). This provides evidence of an independent maturation of the neurohormonal response of hiPSC-CMs in the absence of a sympathetic system and of actual nerve terminations.

## 4. Materials and Methods

### 4.1. Human Adult Cardiomyocytes from Patient Tissue

The study follows the principles of WMA Declaration of Helsinki for medical research involving human subjects. The experimental protocols were approved by the ethical committee of Careggi University-Hospital of Florence (2006/0024713, renewed May 2009; 2013/0035305). The control cohort comprised 10 patients aged <65 years undergoing heart surgery for aortic stenosis or regurgitation and who required a septal myectomy operation due to the presence of a bulging septum causing symptomatic obstruction. All patients had septal thickness <14 mm and preserved left ventricular systolic function (ejection fraction > 55%). Clinical data are found in Supporting Information Appendix A.

### 4.2. Tissue Processing and Cell Isolation

Surgical septal specimens from patients (hAdult-CMs) were washed with standard cardioplegic solution and processed within 30 min from excision. Ventricular tissue was minced and subjected to enzymatic dissociation to obtain viable single myocytes, as previously described [13,31]

### 4.3. HiPSC Cardiac Differentiation and Single Cell Maturation

The isolation of the human cells and the subsequent reprogramming into iPSC lines was performed in conformation with the Declaration of Helsinki. Urine-derived cells from a healthy male donor into hiPSC lines (UC3-4 A1) using a lentiviral vector carrying Oct3/4, Sox2, Klf4 and c-Myc as previously described [32]. For cardiac differentiation we applied a monolayer directed differentiation protocol onto a Matrigel matrix (Matrigel^®^ hESC-Qualified Matrix, Corning^®^, New York, NY, USA) using the cardiac PSC Cardiomyocyte Differentiation Kit (Life Technologies, Thermo Fisher scientific, Carlsbad, CA, USA) following the manufacturer’s instructions as previously described [17]. Briefly, hiPSCs were maintained under feeder-free conditions in mTeSR medium (Stem Cell Technolgies, Vancouver, Canada) on a Corning^®^ Matrigel matrix and regularly passaged every 4–5 days. For cardiac differentiation, hiPSC colonies with 70–80% confluency were chemically dissociated using 1× Tryple (Life Technologies, Thermo Fisher scientific, Carlsbad, CA, USA), suspended into mTeSR with 5μM of ROCK inhibitor (Y27632) and seeded as single cells onto Matrigel-coated wells of a 24 wells plate at a cell density of 40,000 cell/well (24-well plate). At 70% of confluency (2–3 days) the medium is changed to Cardiomyocyte Differentiation Medium A (referred as day 0) to start cardiac induction. Medium A is replaced after two days with Medium B and following other 2 days with Medium C for final differentiation. hiPSC are fed every other day with Medium C until spontaneously beating monolayers appear (day 8–10). At day 12 and for further hiPSC-cardiomyocyte (hiPSC-CM) maturation, Medium C is replaced with RPMI plus B27 supplement (Life Technologies, Thermo Fisher scientific, Carlsbad, CA, USA). On day 20 post differentiation, single cells are obtained from beating monolayers for further cardiac maturation on hydrogel-based micropatterned surfaces.

### 4.4. Maturation on PEG-DA Hydrogel with Micropatterned Topography

Single hiPSC-CMs are obtained from beating monolayers with enzymatic dissociation (10 min at 37 °C) with Tryple (Life Technologies, Thermo Fisher scientific, Carlsbad, CA, USA). Plating media is composed by RPMI/B27 and 10 μM ROCK inhibitor. Cells are seeded at the density of 20,000 cells/cm^2^ onto biomimetic substrates and fed every other day until experimental days. Dual recording experiments of long-term cultured hiPSC-CMs were performed at day 60, 75 and 90 p.d. For patch clamp recordings, hiPSC-CMs were used at day 15, 30 and 60 from cardiac induction.

### 4.5. Fabrication of PEG-DA Hydrogel Substrate with Micropatterned Topography

Micropatterned substrates were prepared by soft lithographic technique. A master sample was replicated by a PDMS (polydimethylsiloxane) mold that is used as template for the pattern replication (Appendix A).

*Master fabrication*—Master samples was obtained by laser writer lithography (μPG101 laser writer, Heidelberg, city, state abbrev, country, 800 nm resolution) (Appendix A). A commercial optical resist (AZ 1505 Merck Performance Materials GmbH, Merck Group, Darmstadt, Germany) was spun over 2 cm × 2 cm Si wafers and exposed to a laser spot (λ = 375 nm) with beam intensity of 16 mW. The length of the linear stripes was set to 1 cm, while their width and spacing were fixed to 0.6 μm and 1.4 μm respectively. After exposure process, the samples were developed for 30 s in a 1:1 solution of AZ Developer (Merck Performance Materials GmbH, Merck Group, Darmstadt, Germany) in water, and subsequently rinsed in deionized water for 120 s.

*Glass slides treatment*—Glasses to support the PEG patterns were silanized to prevent the peeling-off of the hydrogel during the cell culture. First, glasses were washed with an alkaline piranha solution (water, aqueous ammonia and hydrogen peroxide 5:1:1 *v*/*v*) at 70 °C for 15 min. Then, the glass were rinsed with water and the isopropyl alcohol and, after drying, they were immersed in a solution of 3-(trimethoxysilyl) propyl methacrylate) (MAPTMS, 0.064 mM in ethanol) for 1 h. At the end, glasses were washed with isopropanol and dried.

*Fabrication of PDMS mold*—Monomeric PDMS mixture was prepared by mixing the two components of a commercially available PDMS kit (Sylgard 184, Sigma-Aldrich, Merck Group, St. Louis, MI, USA) in a 10:1 *w*/*w* ratio (base and curing agent) and then casted on the silicon master. After curing at 100 °C for 30 min, the crosslinked PDMS mold was peeled off by the master (Appendix A).

*PEG-DA pattern printing*—A small amount (~20 µL) of a solution of PEG-DA (250 Mn, Sigma-Aldrich, Merck Group, St. Louis, MI, USA) and Irgacure 389 photoinitiator (1% *w*/*w*) was dropped on a silanized glass slides and then, the PDMS mold was directly placed onto the surface. Irradiation by UV light (λ = 385 nm)(M385CP1-C4, ThorLabs, Newton, NJ, USA) for 10 min allowed the formation of the crosslinked PEG-DA network. PDMS mold was gently peeled off from the substrates and used again after washing in water and methanol. Images of mold and micropatterned PEG-DA hydrogels were obtained by scanning electron microscopy (SEM) (Appendix A).

### 4.6. Dual Recording of Action Potential and Calcium Transient

For dual recordings of APs and CaTs, hiPSC-CMs were loaded with 2 μL/mL Fluovolt (Thermo Fisher, Waltham, MA, USA), 2 μL/ml of Cal630 (AAT Bioquest, Sunnyvale, CA, USA) and 5 μL of Power Load™ concentrate (Thermo Fisher, Waltham, MA, USA) for 30 min at 37 °C and then washed with pre-warmed culture media before placing the cover slide into the experimental chamber. The experimental chamber features platinum electrodes for electrical field stimulation, connected to a stimulator (DigiTimer, Welwyn Garden City, UK) delivering short (3 ms) voltage pulses. During measurements, cells were continuously perfused with heated Tyrode buffer to keep the temperature stable at 37 ± 1 °C. For fluorescence studies, cells were simultaneously illuminated by LED light at two different wavelengths, blue (488 nm) for excitation of Fluovolt and yellow (580 nm) for Cal630 dye excitation, using a multi-led system (Lumencor SPECTRA X, Beaverton, OR, USA). A dual-wavelenght band-pass filter cube (Semrock, IDEX, Lake Forest, IL, USA) was used to allow fluorescence light from the two dyes to be collected by a single camera (Photometrics Prime sCMOS, Teledyne, Tucson, AZ, USA): in particular, the filter allowed green light (515–545 nm, emission of Fluovolt) and red light (615–655 nm, emission of Cal630) to be collected. In order to separate the two emission wavelengths, we used an OptoSplit II light splitter (Cairn Research Ltd, Kent, UK) that was able to separate the two spectral components of the fluorescence image and focus them simultaneously on the upper and lower half of the camera chip. MetaMorph software (Molecular Devices, San Jose, CA, USA) was used to collect and analyze fluorescence images. The camera collected images at an average rate of 90 frames per second. In each selected microscope view field, a number of single hiPSC-CMs were selected and chosen as regions of interest. The background-corrected average fluorescence values from the pixels in each selected region of interest (myocyte) were recorded at each of the two wavelengths under different stimulation conditions for 5 to 10 s in each condition. For the analysis of action potential or calcium transient kinetics during steady-state stimulation, the average of 5–10 subsequent AP or CaT traces was calculated to reduce noise. Pearson’s correlation coefficient (*r*^2^) was calculated from the linear fitting of values distribution recorded from individual cells.

### 4.7. Perforated Patch Clamp and Calcium Transient Recording

In cardiomyocytes freshly isolated from patient ventricular tissue, we simultaneously measured membrane potential and calcium transients using perforated patch whole-cell current-clamp combined with fluorescence recordings after loading with the Ca^2+^-sensitive fluorescent dye Fluoforte (Enzo Life Sciences, Farmingdale, NY, USA), as previously described [31]: fluorescence was measured at 515 ± 10 nm during excitation at 490 ± 8 nm. For hiPSC-CMs, we used whole-cell current clamp to measure APs; the pipette solution contained (in × 10^−3^ M) 115 K methanesulfonate, 25 KCl, 10 (4-(2-hydroxyethyl)-1-piperazineethanesulfonic acid) (HEPES), 3 MgCl_2_, and cells were perfused with Tyrode buffer containing 1.8 × 10^−3^ M CaCl_2_. APs were elicited with short depolarizing current pulses (< 3 ms) at 1 Hz frequency. Action potentials were analyzed for MDP (mV), amplitude (mV), and action potential duration (ADP50 and APD90, ms) using the Clampfit 10.7 software (Molecular Devices, San Jose, CA, USA).

### 4.8. Statistics

All data are reported as mean ± SEM and were compared using a one-way or two-way analysis of variance (ANOVA). Tukey post hoc test with statistical significance set at * *p* < 0.05 and ** *p*< 0.01 were applied for differences in means between groups/conditions. The interquartile range test was performed for data distribution and selection. For each analysis, *n* represented number of cardiomyocytes and N the total number of cell differentiation runs from individual hiPSC passages (p) or individual patients. For this work healthy cell line had *N* = 3–5. The number of individual experiments (individual coverslides) for each assessment was at least 2 from different differentiation runs. Supporting information of results is reported in Appendix A.

## 5. Conclusions

This is a pilot study on a control hiPSC-CM line that we have widely described before in many aspects [7,9,32]. Although the work is restricted to a single cell line, we aimed to present a simple biomimetic culture approach and a technique for simultaneous comparison of functional output at different maturation time points. In addition, future studies may consider real longitudinal evaluations using genetically-encoded fluorescent voltage or calcium indicators (GEVI or GECI, respectively) to track the functional performance from the same cell at different stages of maturation. The advantage of comparing hiPSC-CMs with adult cardiomyocytes from control patients provided a stronger evidence of the maturation of E-C coupling function towards adult-like ventricular features. However, despite the greater degree of organization and adult-like cell structure, mature hiPSC-CMs still display protein expression and contractile properties that are typically found in the fetal myocardium [7]. For this reason, future directions must provide evidence of how can we overcome the developmental plateau of hiPSC-cardiomyocytes. Functional improvements may come from the alteration of the extracellular milieu towards more physiological conditions or improving multicellular hiPSC-CM tissue models. Finally, for further validation of hiPSC-CM models for personalized-medicine or inherited cardiomyopathy studies, novel insights must be corroborated with clinical information or, ideally, with experiments from cardiomyocytes isolated from the cardiac tissue of the same patient, obtained through biopsies or surgery.

## Figures and Tables

**Figure 1 ijms-20-03799-f001:**
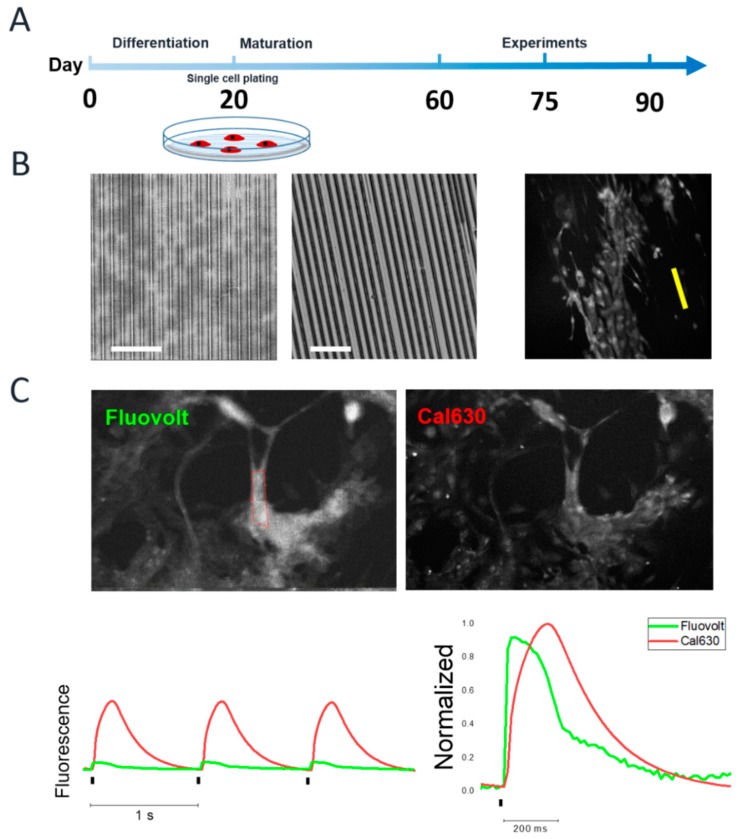
Experimental procedure. Human iPSCs are differentiated into cardiomyocytes (hiPSC-CMs) with monolayer-directed differentiation protocol. (**A**) At day 20 post differentiation, single hiPSC-CMs are seeded onto hydrogel-based micropatterned surface and cultured until experimental day 60, 75 and 90. (**B**) Left: fabrication of micropatterned hydrogel with PDMS mold and PEG-DA hydrogel synthesis by soft lithography (scale bars equal to 10 and 60 μm respectively). hiPSC-CM preferential spreading along thepattern direction (indicated by yellow line). (**C**) Simultaneous recording of action potential and calcium transients using Fluovolt (Ex/em 522/535 nm) and Cal630 (Ex/Em 608/626 nm), respectively.

**Figure 2 ijms-20-03799-f002:**
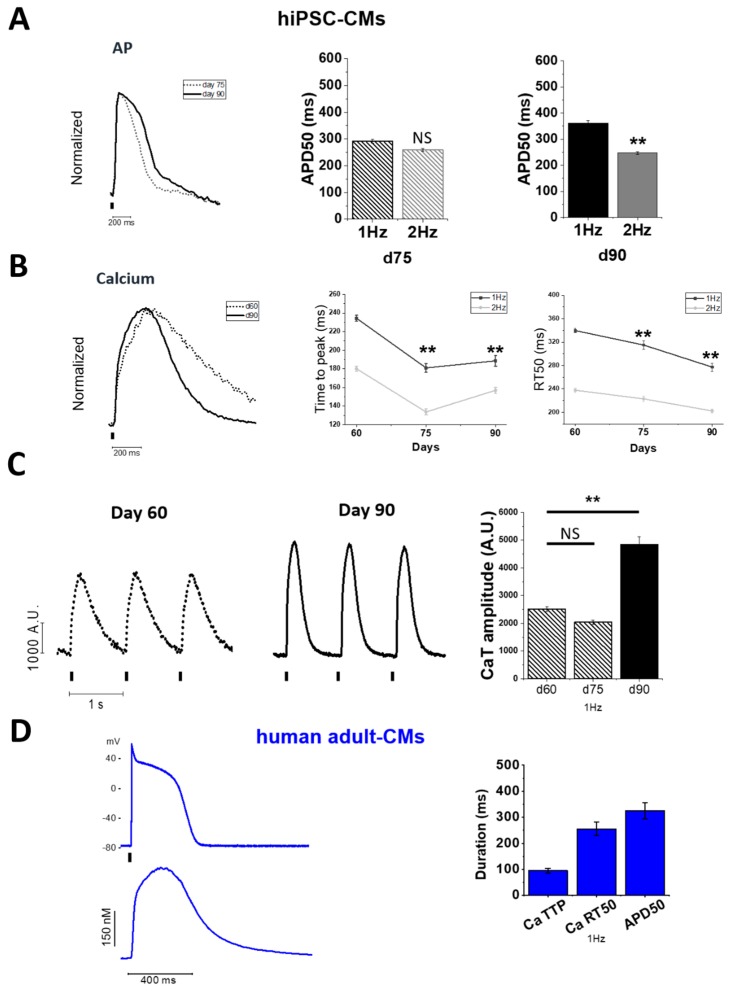
Dual recording of action potential and calcium transient in later-stages hiPSC-CMs. Single hiPSC-CMs matured on hydrogel-based micropatterned surfaces were subjected to simultaneous optical measurements of action potentials and calcium transients under electrical pacing (1 and 2 Hz) at 37 °C at and external [Ca^2+^] = 1.8 mM. (**A**) Superimposed action potential (AP) traces of day 75 (*N* = 2; *n* = 186) vs. day 90 (*N* = 2; *n* = 119) recorded by FluoVolt. AP profile of hiPSC-CMs was recorded both at 1 and 2 Hz to evaluate action potential duration (APD50, ms) and the response to frequency changes at both day 75 and 90. (**B**) Superimposed normalized traces of calcium transients recorded by Cal630 at day 60 (*N* = 3; *n* = 336), 75 (*N* = 5; *n* = 251) and 90 (*N* = 3; *n* = 165): average calcium transient (CaT) rise (time to peak TTP, ms) and CaT decay (difference of 50% of CaT decay and TTP, RT50, ms) are reported, during pacing at 1 and 2 Hz (**C**) Representative CaT profiles at day 60 and 90 and average CaT amplitude (in arbitrary fluorescence units, A.U.) at day 60,75 and 90. (**D**) Representative simultaneous recordings of action potential and intracellular calcium transient from adult ventricular cardiomyocytes, elicited with short current pulses in current-clamp mode at 1 Hz. Average time to peak (Ca TTP) and time from peak to 50% decay (Ca RT50) of Ca transients, and time from stimulus to 50% repolarization (APD50) of action potentials at 1 Hz. Means ± SEM from 27 myocytes (nine control patients). Data are reported as means ± SEM; one-way analysis of variance (ANOVA) with a Tukey post-hoc test with statistical significance set at * *p* < 0.05 and ** *p* < 0.01; NS not significant. Supporting information given in Appendix A. *N* = number of differentiations; *n* = cells.

**Figure 3 ijms-20-03799-f003:**
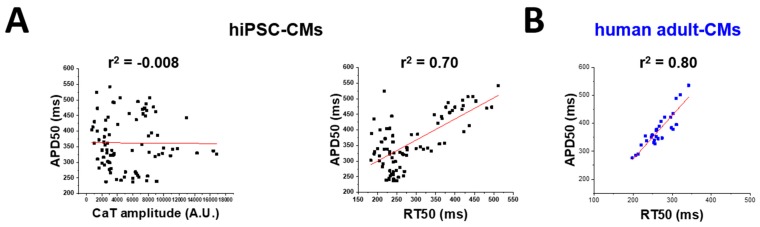
Correlative analysis of action potential and calcium transient parameters. Pearson correlation coefficient (*r*^2^) estimated by linear regression (red line) to correlate APD50 (ms) against CaT amplitude (NS, not significant) and CaT duration (RT50, ms, *p* < 0.05) of (**A**) day 90 hiPSC-CMs and (**B**) (human) hAdult CMs from donor ventricular tissue (*p* < 0.05).

**Figure 4 ijms-20-03799-f004:**
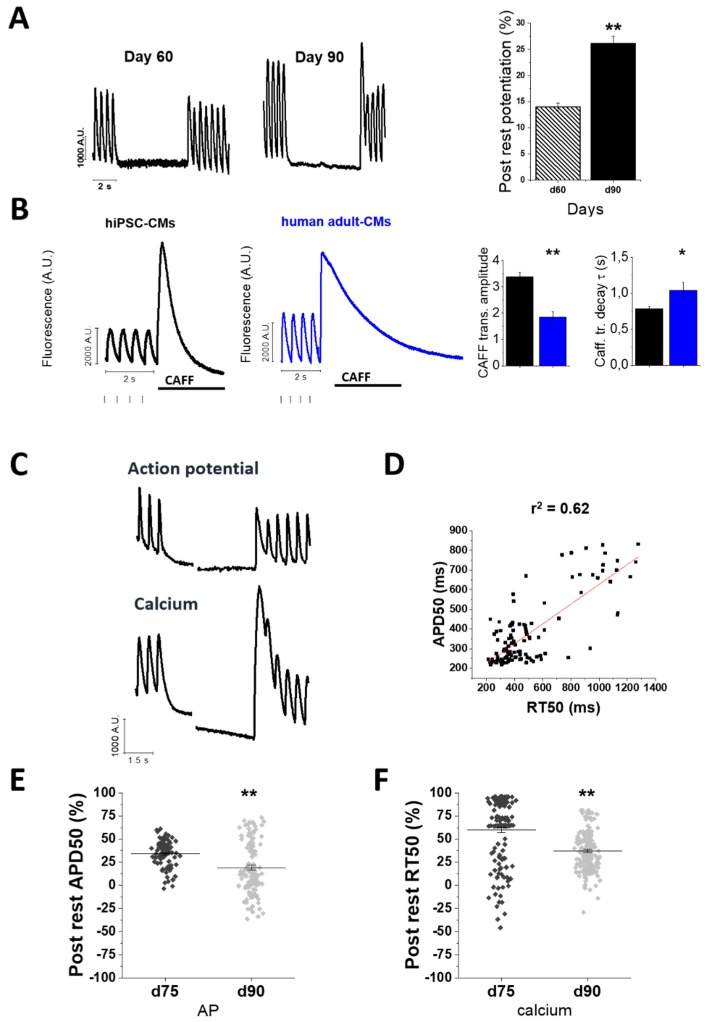
Sarcoplasmic reticulum contribution during hiPSC-CM maturation. Sarcoplasmic reticulum (SR) contribution in calcium handling maturation was tested by a post rest potentiation protocol and caffeine-induced CaTs elicited in hiPSC-CMs at multiple maturation time-points. (**A**) The post-rest potentiation of CaT amplitude was estimated after a resting pause of 5 s, inserted in a regular train of stimulation at 2 Hz. The potentiation is expressed as the % increase of CaT amplitude at the first post-rest beat from that of the last calcium transient before the pause (%). Post rest potentiation is estimated at day 60 and day 90. (**B**) Caffeine-induced CaTs (quick exposure to 10 μM caffeine) after a series of 2 Hz paced CaTs. Average of caffeine transient amplitude was normalized by the amplitude of steady-state calcium transients at 2 Hz prior to caffeine exposure (*N* = 2; *n* = 83). Caffeine transient CaT amplitude (CaT_A CAFF_/CaT_A 2Hz_ ratio) and decay (τ, s^−1^) of hiPSC-CMs were calculated and compared with caffeine-CaT recorded in hAdult-CMs (*N* = 5; *n* = 14). (**C**) Simultaneously recorded APs and CaTs during the pause protocol. (**D**) APs and CaTs from the same cells were compared to show Pearson’s correlation (*r*^2^) between post rest AP duration (APD50, ms) and post rest CaT decay (RT50, ms, *p* < 0.05). (**E**) Variations of post rest APD50 and (**F**) post rest RT50 were measured both at day 75 (AP: *N* = 2, *n* = 119; CaT: *N* = 5, *n* = 251) and day 90 (AP: *N* = 2, *n* = 119; CaT: *N* = 3, *n* = 165). One-way analysis of variance (ANOVA) with a Tukey post-hoc test with statistical significance set at * *p* < 0.05 and ** *p* < 0.01; NS not significant. Supporting information is reported in Appendix A. *N* = number of differentiations or patients; *n* = cells.

**Figure 5 ijms-20-03799-f005:**
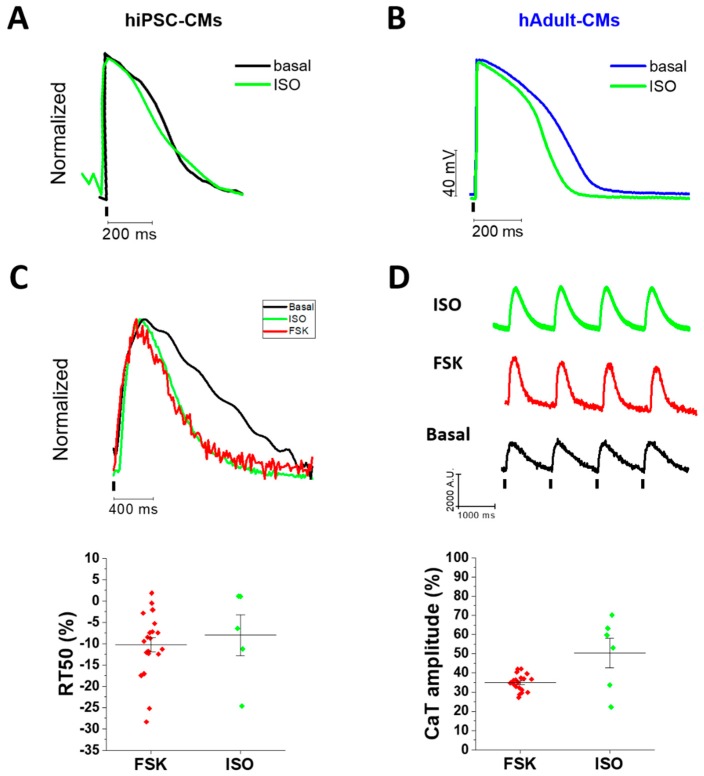
Positive β-adrenergic response of later stage hiPSC-CMs. Day 90 hiPSC-CMs were exposed to isoproterenol (ISO, 1 μM) and forskolin (FSK, 1 μM). (**A**) Representative traces of hiPSC-CM AP recorded before and under ISO stimulation. APD50 reduction under ISO was 12 ± 4%, *p* > 0.05. (**B**) Representative traces of ISO effect on the APD50 of human adult cardiomyocytes (15 ± 3%, *p* < 0.05)(hAdult-CMs: *N* = 5 patients, *n* = 12 cells). (**C**,**D**) Relative positive inotropic and lusitropic effects of both ISO and FSK (%) in later stages hiPSC-CMs (ISO: *N* = 2, *n* = 7; FSK: *N* = 2, *n* = 21). Supporting information is reported in Appendix A.

**Figure 6 ijms-20-03799-f006:**
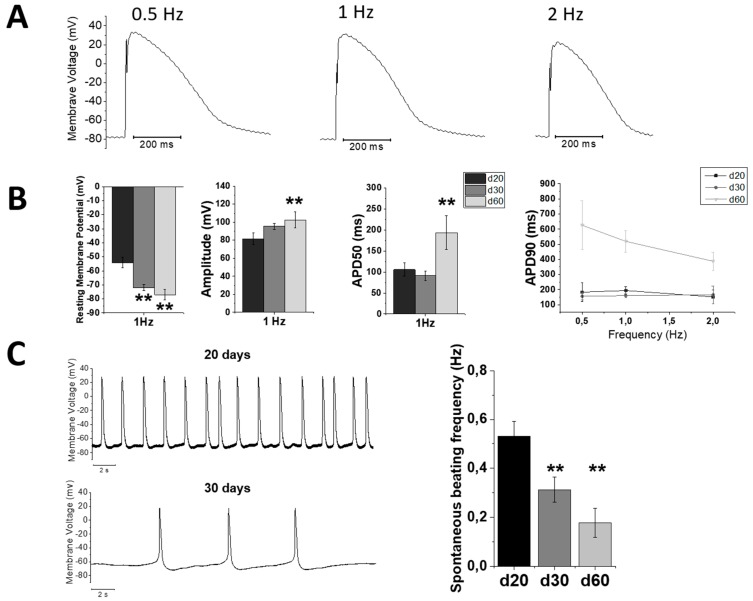
Action potential recording by patch clamp in earlier stages hiPSC-CMs. Earlier-stages hiPSC-CMs during patch clamp, elicited with short current pulses in current-clamp mode at day 20, 30 and 60. (**A**) Representative traces at 0.5, 1 and 2 Hz stimulation rates. (**B**) Average of resting membrane potential (mV), AP amplitude (mV), time from stimulus to 50% repolarization (APD50, ms) at 1 Hz and to 90% of repolarization (APD90, ms) with frequency variation. (**C**) Spontaneous beating frequency of action potential from day 20 to day 60. Data are reported as means ± SEM; One-way analysis of variance (ANOVA) with a Tukey post-hoc test with statistical significance set at * *p* < 0.05 and ** *p* < 0.01; NS not significant.

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
