# Peer review of "Optical Investigation of Action Potential and Calcium Handling Maturation of hiPSC-Cardiomyocytes on Biomimetic Substrates"

_ijms, 2019, doi:10.3390/ijms20153799_

Round 1

Reviewer 1 Report

The authors present an analysis of hiPSC-CM maturation using a simultaneous measurement of calcium and action potential using fluorescent dyes. The paper is generally well written, and the methods are sound. The study provides a number of interesting results concerning calcium handling in developing cardiomyocytes and addresses key issues in hiPSC-CM studies.

However, a few concerns and notes require clarification and attention.

1. The study presents a custom setup for measuring calcium and action potential simultaneously. The study provides a number of valuable results, and a more detailed validation of the setup would give more credit to the study. More numerical data on the traces should be reported – data such as APD50 values with S.E.M. from both patch clamp and optical measurements would be necessary. Currently, these values can only be estimated from the figures.

2. Possible development in parameters that define the relation between the calcium and action potential traces would be of interest. This data would be well accompanied by a figure or video illustrating both signal traces on top of each other – the waveform differences cannot be well discerned from the video supplied in supplementary data.

3. As simultaneous Ca/AP measurements on hiPSC-CMs have been performed during the past few years, I suggest citing related previous research in introduction.

4. In discussion on page 11, starting line 298, you state that the AP durations using Fluovolt was consistent with patch clamp recordings from the same cells. If I understood correctly, patch clamp was measured latest d60 (APD50 near 190 ms according to Figure 6) and Fluovolt earliest d75 (APD50 near 290 ms according to Figure 2). Based on these numbers, the durations would not seem very consistent. Please clarify what you mean by consistency in AP durations here.

5. In Figure 4 and 5 captions, the decreases in amplitude/APD duration were shown as percentages with two decimals. As the frame rate was 90, the number of significant figures seems unlikely to me. Please elaborate, or round the figures appropriately.

In addition, please address the following minor points and typographical errors for consistency:

-In abstract (p. 1 l 29) “faster duration“ likely worded better as “shorter duration”.

-In figure 2E, please indicate the 1 Hz stimulation below the duration bars, as in 2D.

-p.4. l. 119: two dots

-p.8 l 225: 0 in the word post

-p.9 l 246: Figure 5 caption: %-sign missing after ISO reduction.

-p. 9 l 251: double comma

-Please check the manuscript for extra/double spaces in text (several instances).

-Please make sure there is a space between a figure and unit (unit includes %-character).

-p. 11 l 309 missing space

-Error in abbreviation for human adult cardiomyocytes

-p.d. not listed in abbreviations

-CM not listed in abbreviations

Author Response

RESPONSE TO REWIEVER 1

The authors present an analysis of hiPSC-CM maturation using a simultaneous measurement of calcium and action potential using fluorescent dyes. The paper is generally well written, and the methods are sound. The study provides a number of interesting results concerning calcium handling in developing cardiomyocytes and addresses key issues in hiPSC-CM studies.

RESPONSE: We are grateful to the reviewer for the appreciation of the value of our work.

However, a few concerns and notes require clarification and attention.

The study presents a custom setup for measuring calcium and action potential simultaneously. The study provides a number of valuable results, and a more detailed validation of the setup would give more credit to the study. More numerical data on the traces should be reported – data such as APD50 values with S.E.M. from both patch clamp and optical measurements would be necessary. Currently, these values can only be estimated from the figures.

RESPONSE: We thank the reviewer for this suggestion. We have now added a new Supplementary Table 2, in which we clustered all the numerical results as Mean ± SEM.

Possible development in parameters that define the relation between the calcium and action potential traces would be of interest. This data would be well accompanied by a figure or video illustrating both signal traces on top of each other – the waveform differences cannot be well discerned from the video supplied in supplementary data.

RESPONSE: We thank the reviewer for this suggestion. We modified Figure 1C and added a panel with traces from both action potential (Fluovolt) and calcium transients (Cal630) from an individual hiPSC-cardiomyocyte. Normalized traces now show the external pacing that triggers action potential depolarization and, simultaneously, calcium transient rise.

As simultaneous Ca/AP measurements on hiPSC-CMs have been performed during the past few years, I suggest citing related previous research in introduction.

RESPONSE: We partially modified the introduction following the reviewer suggestion. Other studies reporting simultaneous optical recordings of action potential and calcium transients from cell monolayers [Lee P., Circ Res, 2012 doi: 10.1161/CIRCRESAHA.111.262535; REF. 14] or using genetically encoded reporters [Dempsey G.T., J Pharmacol Toxicol Methods, 2016 doi: 10.1016/j.vascn.2016.05.003; REF. 15] at undefined maturation age were cited. Using a similar approach, our dual optical method aimed to systematically define time points of hiPSC-cardiomyocyte maturation of calcium handling and the effect of hydrogel-based micropatterned substrates.

In discussion on page 11, starting line 298, you state that the AP durations using Fluovolt was consistent with patch clamp recordings from the same cells. If I understood correctly, patch clamp was measured latest d60 (APD50 near 190 ms according to Figure 6) and Fluovolt earliest d75 (APD50 near 290 ms according to Figure 2). Based on these numbers, the durations would not seem very consistent. Please clarify what you mean by consistency in AP durations here.

RESPONSE: We agree that the statement was misleading. We corrected as follows: “the duration of APs recorded from individual hiPSC-CMs using the patch-clamp technique (earlier-stages) and Fluovolt approach (later-stages) were consistently prolonged at any progressive time point”

In Figure 4 and 5 captions, the decreases in amplitude/APD duration were shown as percentages with two decimals. As the frame rate was 90, the number of significant figures seems unlikely to me. Please elaborate, or round the figures appropriately.

RESPONSE: We rounded the decimals of the percentage in the figure captions.

In addition, please address the following minor points and typographical errors for consistency:

-In abstract (p. 1 l 29) “faster duration“ likely worded better as “shorter duration”.

RESPONSE: Corrected

 -In figure 2E, please indicate the 1 Hz stimulation below the duration bars, as in 2D.

RESPONSE: Corrected

-p.4. l. 119: two dots.

RESPONSE: Corrected 

-p.8 l 225: 0 in the word post

RESPONSE: Corrected

-p.9 l 246: Figure 5 caption: %-sign missing after ISO reduction.

RESPONSE: Corrected

-p. 9 l 251: double comma

RESPONSE: Corrected

 -Please check the manuscript for extra/double spaces in text (several instances).

RESPONSE: Corrected

 -Please make sure there is a space between a figure and unit (unit includes %-character).

RESPONSE: Corrected

-p. 11 l 309 missing space

RESPONSE: Corrected

 -Error in abbreviation for human adult cardiomyocytes

RESPONSE: Corrected

 -p.d. not listed in abbreviations

RESPONSE: Added

 -CM not listed in abbreviations

RESPONSE: Added

Reviewer 2 Report

The manuscript addresses an important consideration for the rapidly growing collection of investigators using human induced pluripotent stem cell-derived cardiomyocyte aimed at screening new chemical entities in drug discovery and development, investigating cardiac disease and intervention, genetic determinants of cardiac disease, etc.  In the present study, functional properties of hiPSC-CMs were examined on biomimetic substrates and compared to adult cardiomyocytes from fresh ventricular tissue of patients.  It is critical to understand the degree of maturation of hiPSC-CMs to avoid findings that can be translated into misleading conclusions and represents an underappreciated area of investigation.  In the present study, the author compared functional features of hiPSC-CMs and native human ventricular cardiomyocytes freshly isolated from non-failing, non-hypertrophic patients.  They presented a systematic functional characterization of action potential and calcium handling maturation using a dual optical method a various time points in later stages of maturation cultured on biomimetic substrates.  The manuscript is well-written and contributes to the body of research advancing the application of hiPSC-CMs. 

Author Response

RESPONSE TO REVIEWER 2

RESPONSE: We sincerely thank the reviewer for the deliberate comments on our work.